# Self-Learning Transformations for Improving Gaze and Head Redirection

**Yufeng Zheng**[1], **Seonwook Park**[1], **Xucong Zhang**[1], **Shalini De Mello**[2], **Otmar Hilliges**[1]

[1]Department of Computer Science, ETH Zurich      [2]NVIDIA

{firstname.lastname}@inf.ethz.ch    shalinig@nvidia.com

## Abstract

Many computer vision tasks rely on labeled data. Rapid progress in generative modeling has led to the ability to synthesize photorealistic images. However, controlling specific aspects of the generation process such that the data can be used for supervision of downstream tasks remains challenging. In this paper we propose a novel generative model for images of faces, that is capable of producing high-quality images under fine-grained control over eye gaze and head orientation angles. This requires the disentangling of many appearance related factors including gaze and head orientation but also lighting, hue etc. We propose a novel architecture which learns to discover, disentangle and encode these extraneous variations in a self-learned manner. We further show that explicitly disentangling task-irrelevant factors results in more accurate modelling of gaze and head orientation. A novel evaluation scheme shows that our method improves upon the state-of-the-art in redirection accuracy and disentanglement between gaze direction and head orientation changes. Furthermore, we show that in the presence of limited amounts of real-world training data, our method allows for improvements in the downstream task of semi-supervised cross-dataset gaze estimation. Please check our project page at: https://ait.ethz.ch/projects/2020/STED-gaze/

## 1 Introduction

Extracting information from images of human faces is one of the core problems in artificial intelligence and computer vision. For example, estimating eye gaze has many applications in the social sciences [1, 2], cognitive sciences [3, 4], and can enable novel applications in graphics, HCI, AR and VR [5, 6, 7, 8, 9, 10]. Given the need for large amounts of training data for learning-based gaze estimation approaches, much attention has been given to synthesizing training data via a graphics pipeline [11, 12] and synthetic-to-real domain adaptation [13]. However, domain adaptation approaches can be sensitive to changes in the underlying distribution of gaze directions, producing unfaithful images that do not help in improving gaze estimator performance [14]. More recently, approaches to re-direct the gaze in real-images has emerged as an alternative approach to attain gaze estimation training data [15, 16, 17, 12]. This task involves the learning of a mapping between two images with differing gaze directions and requires either paired synthetic [16] or high-quality real-world images captured under controlled conditions [17].

However, leveraging gaze data from in-the-wild settings [18, 19] for this purpose has so far been elusive [15, 20]. The underlying factors that we wish to explicitly control (gaze, head orientation) are entangled with many other extraneous factors (e.g., lighting, hue, etc) that are typically not known *a priori* (see Fig. 2, first and last columns). Conditional unpaired image-to-image translation methods such as StarGAN [21] and similar [22, 23, 24] provide a promising framework to tackle this in-the-wild problem, where perfectly paired images are not available. However, accurately controlling the explicit factors to a degree where the generated data is useful for downstream tasks remains challenging.

To solve this challenge we propose a new approach for gaze and head orientation redirection along with a principled way to evaluate such methods, and demonstrate its effectiveness at improving the challenging downstream tasks of semi-supervised cross-dataset gaze estimation. Specifically, we design a novel framework that learns to simultaneously encode *both* the explicit task-relevant (gaze, head orientation) and the extraneous, unlabeled task-irrelevant factors. To do this, we propose a self-transforming encoder-decoder architecture with multiple transformable factors at the bottleneck. Each factor consists of a latent embedding and a self-predicted pseudo condition, which describes the amount of its variation as present in a particular image. In addition, we propose several novel constraints to effectively disentangle the various independent factors, while maintaining precise control over the explicitly manipulated factors in the redirected images.

We introduce several evaluation schemes to assess the quality of redirection in a principled manner. First, we propose a *redirection error* metric which measures how accurately the target gaze direction and head orientation values are reproduced in the generated images. Second, we propose a *task disentanglement error* metric which measures how much the perceived gaze direction or head orientation changes when other factors are adjusted. With these two metrics, we show that our novel architecture is effective in isolating a multitude of independent factors, and thus performs well in terms of faithfulness of redirection. To further evaluate the impact of our proposed approaches, we demonstrate the effectiveness of our gaze redirection scheme on the task of semi-supervised cross-dataset gaze estimation. Here, we first train our gaze redirection network in a semi-supervised manner with a small amount of labeled training data. Then we augment the labeled data via redirection and train a gaze network on the augmented data to demonstrate improvements compared to training on only the non-augmented labeled data.

In summary, we contribute:

- a novel self-transforming encoder-decoder architecture that learns to control both explicit and extraneous factors in a disentangled manner,
- a principled evaluation scheme for measuring the accuracy of gaze redirection methods, and the disentanglement between task-specific (explicit) and task-irrelevant (extraneous) factors,
- high-fidelity gaze and head orientation manipulation on the generated face images, and
- demonstration of performance improvements on four datasets in the real-world downstream tasks of cross-dataset gaze estimation, by augmenting real training data via redirection.

## 2 Background

**Gaze Redirection.** DeepWarp is an early work that learns a warping map between pairs of eye images with different gaze directions [15]. It only works for a limited range of target gaze directions. Yu *et al.* extend it with synthetic data from [25] along with a gaze estimator [16]. Warping-based methods cannot generate new image content since every pixel in the generated image is interpolated from the original input image. Therefore, such methods cannot synthesize the change of lighting conditions or extreme gaze directions and head orientations. He *et al.* present a GAN-based architecture where perceptual loss [26] and cycle consistency [27] are used to supervise the redirection process. However, their work uses high-resolution images with controlled lighting conditions and cannot generalize well to in-the-wild images [17]. Alternatively, Wood *et al.* [28] fit morphable models to eye regions to redirect them, but are limited by the fidelity and flexibility of the morphable model. FAZE uses a disentangling encoder-decoder architecture to learn a latent representation to rotate gaze and head orientation, but its redirection yields images of low quality [20]. Unfortunately, previous methods only work with eye region inputs, requiring high-quality images for training, and suffer from a lack of fidelity in preserving gaze in many cases. We advance this task by generating high-fidelity face images with target gaze and head orientation along with control over many other independent factors.

**Cross-Dataset Gaze Estimation.** Cross-dataset gaze estimation involves training and testing on different datasets and is a long-standing unsolved problem [29]. Shrivastava et al. propose to use a GAN for unsupervised domain adaptation of synthetic data [13]. Wood *et al.* synthesize large numbers of eye images as training data for a k-NN model and test it on other datasets [12]. Representations of the internal geometry of the problem have helped with this too, with the 3D eyeball model used in [30] or landmarks in [31]. However, the lack of a means to model the distribution of the properties of a target test dataset remains a key challenge. Our work is a meaningful step towards generating realistic images with controllable properties for cross-domain generic regression training.

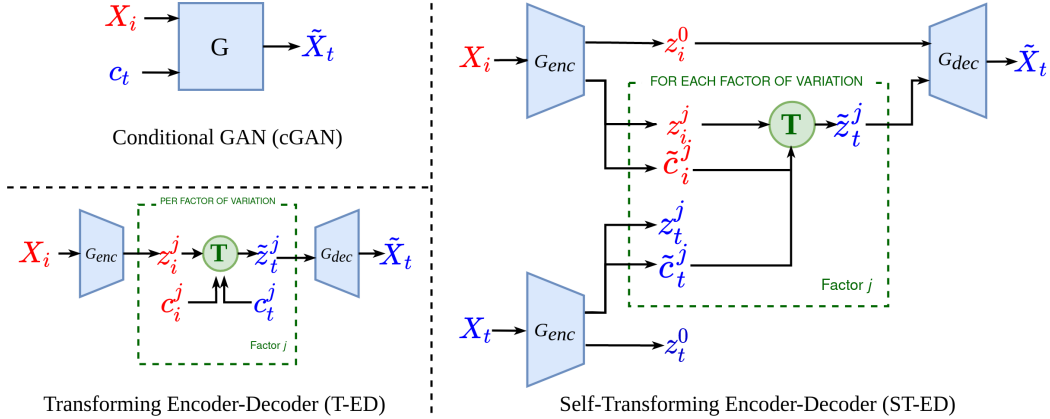

Figure 1: **Conceptual Overview.** Our proposed Self-Transforming Encoder-Decoders (ST-ED) can reduce reliance on noisy labels ($c_i$ and $c_t$) in the conditional image-to-image translation task: $(X_i, \boldsymbol{c_t}) \rightarrow X_t$. This is because the conditions used as input to the transformations are predicted as *pseudo labels* ($\tilde{c}_i$ and $\tilde{c}_t$) and can deviate from the labels used for supervision. This also allows for extraneous factors to be learned without supervision on the predicted conditions.

**Fine-grained Conditional Image Generation.** Conditional image generation takes as input an image and a condition, and generates a transformed image which reflects the condition while preserving other aspects of the input image. Many works achieve photo-realism in conditional image generation by carefully designing the flow of content and style information [32, 33, 34, 35, 36]. However, fine-grained control of conditions is more commonly performed by taking the target condition directly as input into a conditional GAN [37] as shown in StarGAN [21] and its derivatives [22, 23, 24]. For fine-grained control of continuous-valued conditions, transforming architectures, e.g., [38] are particularly effective. On synthetic data, transforming architectures are demonstrated to interpolate precisely in a measurable way [39, 40] or qualitatively [41], and show promising extrapolation possibilities [42]. These approaches are also effective in real-world downstream tasks such as semi-supervised human pose estimation [43] and few-shot gaze estimator adaptation [20], and thus are highly applicable to our task setting. We demonstrate a novel transforming architecture in this paper, which not only produces photo-realistic outputs despite noisy real-world training data, but also better reflects target gaze and head orientation conditions while disentangling extraneous factors.

## 3 Method

### 3.1 Problem Setting

Our goal is to train a conditional generative network, such that given an input image $X_i$ and a set of target conditions $\boldsymbol{c_t}$, it generates an output image $X_t$ by learning the mapping: $(X_i, \boldsymbol{c_t}) \rightarrow X_t$ (Fig. 1 top-left). Following previous works [42, 20], we use a pair of images $X_i$ and $X_t$ of one person as input during training. We assume that each image can be represented by a personal non-varying embedding $z^0$ along with $N$ factors of image variations $\boldsymbol{f} = \{f^1, f^2, ..., f^N\}$. Each factor $f_i^j$ of an image $X_i$ can be described by an embedding $z_i^j$ and a continuous real-valued condition $c_i^j$, which describes the amount of deviation from the canonical state. Each condition $c_i^j$ is an $n$-dimensional vector where $n$ represents the degrees of freedom for the factor. Hence, $X_i$'s overall image condition is denoted by $\boldsymbol{c_i} = \{c_i^1, c_i^2, ..., c_i^N\}$. We further assume that a subset of the factors are "extraneous" to the task ($\boldsymbol{f_u}$ with conditions $\boldsymbol{c_u}$) and are unknown *a priori* (e.g., environmental lighting, hue, shadow, blurriness, and camera distance), while other "explicit" factors are known (e.g., $c^g$ and $c^h$ for gaze and head orientation, respectively) and labeled with ground truth values.

To solve this, we propose a novel architecture based on the concept of transforming encoder-decoders (T-ED), where labeled factors of image variation are learned via rotationally equivariant mappings[1] between independent latent embedding spaces and the image space (Fig. 1 bottom-left). We extend T-ED to additionally discover, encode and disentangle multiple extraneous unknown factors in a self-supervised manner. We call our architecture the Self-Transforming Encoder-Decoder (ST-ED, Fig. 1 right). For a pair of images $X_i$ and $X_t$, ST-ED *predicts* their personal non-varying codes ($z_i^0$ and $z_t^0$), their pseudo-label conditions ($\tilde{\boldsymbol{c}}_i$ and $\tilde{\boldsymbol{c}}_t$) and embedding representations ($\boldsymbol{z}_i$ and $\boldsymbol{z}_t$). Transforming with pseudo condition labels at training time, allows for the discovery of unknown extraneous factors in the absence of ground-truth. In addition, pseudo-label predictions reduce reliance on noisy task-specific labels (typically used as input conditions in prior works) by performing the transformations with conditions that should better match the expected target image $X_t$. At inference time, we replace the pseudo target conditions $\tilde{\boldsymbol{c}}_t$ with the desired target conditions $\boldsymbol{c}_t$.

## 3.2  Model overview

Our redirection network $G$ consists of an encoder $G_{enc}$ and a decoder $G_{dec}$ (see Fig. 1 right). $G_{enc}$ estimates the factors $\boldsymbol{f}$ and the personal non-varying embedding $z^0$ of an image. To begin the redirection process, we first encode both the input and target images by:
$$\{z_i^0, f_i^1, f_i^2, ..., f_i^N\} = G_{enc}(X_i), \quad \{z_t^0, f_t^1, f_t^2, ..., f_t^N\} = G_{enc}(X_t), \qquad (1)$$
where $f_i^j = \{z_i^j, \tilde{c}_i^j\}$ and $f_t^j = \{z_t^j, \tilde{c}_t^j\}$ are the predicted factors from the input and target images, respectively. We regard the condition $\tilde{c}^j$ predicted by $G_{enc}$ for the factor $f^j$ as its pseudo label. Specifically in our setting, we define $f^g$ and $f^h$ as explicit factors corresponding to gaze and head orientation, and supervise their pseudo conditions with ground-truth labels $\boldsymbol{c} = \{c^g, c^h\}$ (see Sec. 3.3). We transform each input embedding $z_i^j$ with the pseudo label conditions $\tilde{c}_i^j$ and $\tilde{c}_t^j$ for the input and target images:
$$\tilde{z}_t^j = T(z_i^j, \tilde{c}_i^j, \tilde{c}_t^j) = \boldsymbol{R}_t^j \left(\boldsymbol{R}_i^j\right)^{-1} z_i^j, \qquad j \in \{1, 2, ..., N\} \qquad (2)$$
where $\boldsymbol{R}_i^j$ and $\boldsymbol{R}_t^j$ are rotation matrices computed from $\tilde{c}_i^j$ and $\tilde{c}_t^j$, respectively. The transformation $T(\cdot)$ first rotates the input embedding $z_i^j$ back to a canonical representation and then to the target condition (see supplementary materials for more details). Intuitively, the canonical orientation of gaze and head orientation would align with explicitly defined spherical coordinate systems (where the zero-configuration points towards the camera), whereas for extraneous factors such as directional lighting, the empirical mean observation of the training dataset may become associated with the canonical form. The transformed embedding can be represented by $\tilde{\boldsymbol{z}}_t = \{z_i^0, \tilde{z}_t^1, \tilde{z}_t^2, ..., \tilde{z}_t^N\}$. Note that $z_t^0$ is directly passed through, assuming that for a given person (from whose data we sample $X_i$ and $X_t$) there exist unique non-varying factors that define their appearance. Finally, we decode the transformed embeddings to generate the redirected image with $\tilde{X}_t = G_{dec}(\tilde{\boldsymbol{z}}_t)$.

## 3.3  Objectives

**Reconstruction Loss.** We guide the generation of redirected images with a pixel-wise $L_1$ reconstruction loss between the generated image $\tilde{X}_t$ and the target image $X_t$:
$$\mathcal{L}_R(\tilde{X}_t, X_t) = \frac{1}{|X_t|} \left\| \tilde{X}_t - X_t \right\|_1. \qquad (3)$$
**Functional Loss.** Inspired by the perceptual loss [17, 26, 44], we propose a novel functional loss which prioritizes the minimization of task-relevant inconsistencies between the generated and target images, e.g., the mismatch in iris positions for our case. Analogous to its original form, we assess the generated image $\tilde{X}_t$ and target image $X_t$ with a feature-consistency loss which is formulated as the $L_2$ loss between the feature maps of $\tilde{X}_t$ and $X_t$:
$$\mathcal{L}_{F_{feature}}(\tilde{X}_t, X_t) = \sum_{i=1}^{5} \frac{1}{|\psi_i(X_t)|} \left\| \psi_i(\tilde{X}_t) - \psi_i(X_t) \right\|_2, \qquad (4)$$
where $\psi_i(\cdot)$ calculates the activation feature maps of the $i$-th layer of a network $F_d$, which is pre-trained on specific tasks (in our case, gaze direction and head orientation estimation) in order to extract task-specific features. We additionally use a content-consistency loss which is formulated as the angular error between the predicted gaze direction and head orientation values from $\tilde{X}_t$ and $X_t$:
$$\mathcal{L}_{F_{content}}(\tilde{X}_t, X_t) = \mathcal{E}_{ang}\left(F_d^g\left(\tilde{X}_t\right), F_d^g\left(X_t\right)\right) + \mathcal{E}_{ang}\left(F_d^h\left(\tilde{X}_t\right), F_d^h\left(X_t\right)\right), \qquad (5)$$

$$\mathcal{E}_{\text{ang}}(\mathbf{v}, \hat{\mathbf{v}}) = \arccos \frac{\mathbf{v} \cdot \hat{\mathbf{v}}}{\|\mathbf{v}\| \|\hat{\mathbf{v}}\|}. \tag{6}$$

Note that instead of directly using the ground-truth labels of the target image, we assess the consistency of labels predicted by $F_d$. We hypothesize that this mitigates the deviation caused by the systematic error of $F_d$ and better aligns with our reconstruction objective. Our final functional loss is defined as:

$$\mathcal{L}_{\text{F}} = \lambda_{F_{feature}} \mathcal{L}_{\text{F}_{\text{feature}}} + \mathcal{L}_{\text{F}_{\text{content}}}. \tag{7}$$

**Disentanglement Loss.** Individual factors should ideally be disentangled such that changing a subset of factors would not alter any of the other factors in the generated image $\tilde{X}_t$. We encourage this disentanglement among the encoded factors, by first randomly transforming a subset of factors to create a mixed factor representation. This is formulated as:

$$\boldsymbol{f_{mix}} = \{f_{mix}^1, f_{mix}^2, ..., f_{mix}^N\}, f_{mix}^j = s f_i^j + (1-s) \tilde{f}_t^j, \quad s \sim \{0, 1\}. \tag{8}$$

We decode these mixed embeddings and encode the synthesized image back to factors $\boldsymbol{f_{rec}} = G_{enc}(G_{dec}(\boldsymbol{z_{mix}}))$. We represent the full disentanglement loss by the discrepancy between the mixed and recovered embeddings, $\boldsymbol{z_{mix}}$ and $\boldsymbol{z_{rec}}$, respectively, as well as by the error between the gaze and head labels before and after the process:

$$\mathcal{L}_{\text{D}} = \mathcal{E}_{\text{ang}}(\tilde{c}_{mix}^g, \tilde{c}_{rec}^g) + \mathcal{E}_{\text{ang}}(\tilde{c}_{mix}^h, \tilde{c}_{rec}^h) + \left(1 - \frac{\boldsymbol{z_{mix}} \cdot \boldsymbol{z_{rec}}}{\|\boldsymbol{z_{mix}}\| \|\boldsymbol{z_{rec}}\|}\right). \tag{9}$$

**Explicit Pseudo-Labels Loss.** In our generator, two of our defined factors correspond to those for which we have explicit ground-truth labels, namely the gaze direction and head orientation. Knowing this, the set of all pseudo labels $\tilde{c}$ can be written as $\tilde{\boldsymbol{c}} = \{\tilde{c}^{u,1}, \tilde{c}^{u,2}, ..., \tilde{c}^{u,N-2}, \tilde{c}^g, \tilde{c}^h\}$, where $\tilde{c}^{u,j}$ represents the pseudo label for the $j$-th factor without available ground-truth (extraneous factors), and $\tilde{c}^g$ and $\tilde{c}^h$ are the pseudo labels for the gaze direction and head orientation (explicit factors). The corresponding ground-truth values are defined as $\{c^g, c^h\}$. To guide the learning of the explicit factors, we use an estimation loss on the gaze direction and head orientation pseudo labels:

$$\mathcal{L}_{\text{PL}} = \mathcal{E}_{\text{ang}}(c^g, \tilde{c}^g) + \mathcal{E}_{\text{ang}}(c^h, \tilde{c}^h). \tag{10}$$

**Embedding-Consistency Loss.** As $X_i$ and $X_t$ are sampled from the same person's data, it is expected that the latent embeddings from different samples of the same person would be consistently learned. However, as shown in [20], a loss term for maintaining consistency helps in preserving person-specific information. Therefore, we use a consistency loss between the transformed input embeddings $\tilde{z}_t$ and the target embeddings $z_t$, after flattening both embeddings:

$$\mathcal{L}_{\text{EC}} = 1 - \frac{\tilde{z}_t \cdot z_t}{\|\tilde{z}_t\| \|z_t\|}. \tag{11}$$

**GAN Loss.** We use a standard GAN loss [45] to encourage photo-realistic outputs from the generator $G$. We choose PatchGAN [46, 13] as our discriminator $D$ and use:

$$\mathcal{L}_{\text{discriminator}}(G, D) = \mathbb{E}\left[\log D\left(X_t\right) + \log\left(1 - D\left(\tilde{X}_t\right)\right)\right], \tag{12}$$

$$\mathcal{L}_{\text{generator}}(D) = \mathbb{E}\left[\log D\left(\tilde{X}_t\right)\right]. \tag{13}$$

**Full Loss.** The combined loss function for the training of the generator is:

$$\mathcal{L}_{\text{full}} = \lambda_R \mathcal{L}_{\text{R}} + \lambda_F \mathcal{L}_{\text{F}} + \lambda_{PL} \mathcal{L}_{\text{PL}} + \lambda_{EC} \mathcal{L}_{\text{EC}} + \lambda_D \mathcal{L}_{\text{D}} + \mathcal{L}_{\text{generator}}, \tag{14}$$

where we empirically set $\lambda_R = 200, \lambda_F = 20, \lambda_{PL} = 5, \lambda_{EC} = 2$, and $\lambda_D = 2$.

## 4 Results

### 4.1 Implementation details

We parameterize $G_{enc}$ and $G_{dec}$ with a DenseNet-based architecture as done in [20]. We implement the external gaze direction and head orientation estimation network $F_d$ by a VGG-16 [47] based architecture which outputs its predictions in spherical coordinates [29]. For evaluation, we use a separate estimator $F_d'$ network that is based on ResNet-50 [48] and is unseen during training, though trained on the same training data. We calculate ground truth labels $\{c^g, c^h\}$ via the data normalization procedure [49, 50] used to pre-process gaze datasets where head orientation is defined without the roll component. Further implementation details are in our supplementary materials.

We train ST-ED using the GazeCapture training subset [18], the largest publicly available gaze dataset. It consists of 1474 participants and over two million frames taken in unconstrained settings, which makes it challenging to train with. As such, to the best of our knowledge, we are the first to demonstrate that photo-realistic gaze redirection models can be learned from such noisy data. To evaluate our models, we use GazeCapture (test subset), MPIIFaceGaze [51], Columbia Gaze [8], and EYEDIAP [52]. Each dataset exhibits different distributions of head orientation and gaze direction, as well as differences in the present extraneous factors. This cross-dataset experiment allows for a better characterization of our approach in comparison to the state-of-the-art approaches.

Table 1: **Ablation study (lower is better).** Our FAZE [20]-like T-ED base model learns only explicit factors (gaze direction and head orientation). We found that controlling these explicit factors with self-predicted pseudo labels (ST-ED Base Model) helps to improve redirection and disentanglement scores. Building on the ST-ED architecture, we allow for extraneous factors $f_u$ to be discovered which further boosts performance. Additionally, our novel functional ($\mathcal{L}_F$) and disentanglement ($\mathcal{L}_D$) losses enforce accurate generation of task-relevant features and clean disentanglement between different factors, improving all metrics. Each of the changes listed in the first column are with respect to the immediate previous row.

| Approach | Gaze Direction | | | Head Orientation | | | LPIPS | |
|---|---|---|---|---|---|---|---|---|
| | Re-dir. | $u \to g$ | $h \to g$ | Re-dir. | $u \to h$ | $g \to h$ | $g + h$ | all |
| T-ED Base Model [20] | 7.114 | - | 4.882 | 2.470 | - | 0.542 | 0.279 | 0.279 |
| ST-ED Model | 5.107 | - | 3.639 | 1.479 | - | 0.660 | 0.272 | 0.271 |
| $+f_u$ | 4.716 | 0.814 | 3.404 | 1.434 | 0.314 | **0.385** | 0.257 | 0.215 |
| $+\mathcal{L}_F + \mathcal{L}_D$ | **2.195** | **0.507** | **2.072** | **0.816** | **0.211** | 0.388 | **0.248** | **0.205** |

## 4.2 Evaluation Metrics

We evaluate our gaze redirector with three kinds of measurements: redirection error, disentanglement of factors, and perceptual quality. All metrics are better when lower in value.

**Redirection Error.** We quantify the fulfillment of the explicit conditions in our image outputs by assessing gaze and head orientation with an external ResNet-50 based [48] estimator $F'_d$ that is unseen during training. We report the redirection error as the angular error between the estimated values from $F'_d$ and their intended target values.

**Task Disentanglement Error.** In contrast to generalized disentanglement metrics such as the $\beta$-VAE metric [53], we directly measure the effect of other factors on the gaze and head orientation factors. To measure the effect of a factor $f^j$ on a factor $f^k, k \in \{g, h\}$, we first randomly perturb the condition $\tilde{c}^j$ with noise sampled from a uniform distribution, $\varepsilon \sim \mathcal{U}(-\eta, \eta)$, such that the perturbed condition is $\tilde{c}^{j'} = \tilde{c}^j + \varepsilon$. We choose $\eta = 0.1\pi$ in our evaluations. We then transform the associated embedding with $z'_j = T(z_j, \tilde{c}_j, \tilde{c}'_j)$ and decode the new embeddings via $\bar{X}' = G_{dec}(z')$. We apply the gaze and head orientation estimator $F'_d$ on the *perturbed* $\bar{X}'$ and unperturbed $\tilde{X}$ reconstructions. We then compare the predicted gaze and head orientation directions from the two images using the error function $\mathcal{E}_{ang}$ (see Eq. 6). To perturb multiple factors, we simply take the average of all calculated errors over the number of factors perturbed. We specifically evaluate (a) $u \to g$, the effect of changes in all extraneous factors on apparent gaze direction, (b) $u \to h$, the effect of changes in all extraneous factors on apparent head orientation, (c) $g \to h$, the effect of changes in the gaze direction factor on apparent head orientation and vice versa ((d) $h \to g$).

**LPIPS.** The Learned Perceptual Image Patch Similarity (LPIPS) metric [54] was previously used by [17] to measure paired image similarity in gaze redirection, and thus we adopt it here. Our ST-ED architecture is able to align to all predicted conditions from the target image (*all* in Tab. 1) or just the gaze direction and head orientation conditions ($g + h$ in Tab. 1) and as such we report both scores.

## 4.3 Ablation study

We report results of an ablation study evaluated on the GazeCapture test set [18] in Table 1, where we show the three metrics. Our base model uses a transforming encoder-decoder (T-ED) architecture with only gaze direction and head orientation factors, which is similar to FAZE [20]. We train our T-ED base model with reconstruction $\mathcal{L}_R$, adversarial $\mathcal{L}_{generator}$ and embedding consistency $\mathcal{L}_{EC}$ losses. Our ST-ED model in the second row of Table 1 additionally predicts gaze and head pseudo labels which are used for transforming the corresponding factors. We can see that controlling the explicit transformations with self-predicted pseudo labels during training helps to improve the redirection and disentanglement scores. This is because self-prediction of pseudo-labels helps to reduce the harm of confounding and noisy labels. From the third row of Table 1, we can see that learning to discover and disentangle the extraneous factors via our framework improves redirection, disentanglement and LPIPS scores further. When using only explicit, task-specific factors, the model is forced to embed extraneous changes into the gaze and head orientation factors in order to satisfy reconstruction-related losses, thus deteriorating redirection accuracy in outputs. We also see that when the extraneous factors are aligned to predicted conditions in the target image, the LPIPS score improves (all vs

$g + h$). Qualitatively, we can see in Fig. 2e that our approach can align better to a given target image on additionally transforming our extraneous factors, compared to when only redirecting the explicit factors (Fig. 2d). When we add the functional $\mathcal{L}_F$ and disentanglement $\mathcal{L}_D$ losses (row 4), they yield large and consistent improvements in all error metrics. The functional loss encourages accurate reconstruction of task-relevant features, while the disentanglement loss enforces that explicit variations are only influenced by the corresponding factors of ST-ED. Our results indicate that the disentanglement of factors and the reconstruction of task-related conditions are aligned objectives.

## 4.4 Comparisons to the state-of-the-art baselines

Table 2: **State-of-the-art comparisons.** We compare our best model against StarGAN [21] and He *et al.* [17] on the task of full-face gaze and head redirection, evaluated on four gaze datasets. Our approach not only generates gaze direction and head orientation more faithfully, but also achieves better disentanglement for separately controlling the two properties. Furthermore, our model allows for the manipulation of extraneous factors, enabling us to out-perform in terms of perceptual image quality as well (for the row *Ours*, we calculate LPIPS after aligning all factors to a target image using its pseudo-labels $\tilde{c}_t$).

<table>
<tr><td colspan="6" align="center">(a) GazeCapture</td></tr>
<tr><td></td><td>Gaze Redir.</td><td>Head Redir.</td><td>$g \rightarrow h$</td><td>$h \rightarrow g$</td><td>LPIPS</td></tr>
<tr><td>StarGAN</td><td>4.602</td><td>3.989</td><td>0.755</td><td>3.067</td><td>0.257</td></tr>
<tr><td>He <em>et al.</em></td><td>4.617</td><td>1.392</td><td>0.560</td><td>3.925</td><td>0.223</td></tr>
<tr><td>Ours</td><td><strong>2.195</strong></td><td><strong>0.816</strong></td><td><strong>0.388</strong></td><td><strong>2.072</strong></td><td><strong>0.205</strong></td></tr>
</table>

<table>
<tr><td colspan="6" align="center">(b) MPIIFaceGaze</td></tr>
<tr><td></td><td>Gaze Redir.</td><td>Head Redir.</td><td>$g \rightarrow h$</td><td>$h \rightarrow g$</td><td>LPIPS</td></tr>
<tr><td>StarGAN</td><td>4.488</td><td>3.031</td><td>0.786</td><td>2.783</td><td>0.260</td></tr>
<tr><td>He <em>et al.</em></td><td>5.092</td><td>1.372</td><td>0.684</td><td>3.411</td><td>0.241</td></tr>
<tr><td>Ours</td><td><strong>2.233</strong></td><td><strong>0.884</strong></td><td><strong>0.365</strong></td><td><strong>1.849</strong></td><td><strong>0.203</strong></td></tr>
</table>

<table>
<tr><td colspan="6" align="center">(c) Columbia</td></tr>
<tr><td></td><td>Gaze Redir.</td><td>Head Redir.</td><td>$g \rightarrow h$</td><td>$h \rightarrow g$</td><td>LPIPS</td></tr>
<tr><td>StarGAN</td><td>6.522</td><td>3.444</td><td>1.029</td><td>3.359</td><td>0.255</td></tr>
<tr><td>He <em>et al.</em></td><td>7.345</td><td>1.677</td><td>0.692</td><td>3.831</td><td><strong>0.227</strong></td></tr>
<tr><td>Ours</td><td><strong>3.333</strong></td><td><strong>1.095</strong></td><td><strong>0.452</strong></td><td><strong>2.136</strong></td><td>0.242</td></tr>
</table>

<table>
<tr><td colspan="6" align="center">(d) EYEDIAP</td></tr>
<tr><td></td><td>Gaze Redir.</td><td>Head Redir.</td><td>$g \rightarrow h$</td><td>$h \rightarrow g$</td><td>LPIPS</td></tr>
<tr><td>StarGAN</td><td>14.906</td><td>3.929</td><td>0.915</td><td>4.025</td><td>0.248</td></tr>
<tr><td>He <em>et al.</em></td><td>13.548</td><td>1.581</td><td>0.663</td><td>4.367</td><td>0.218</td></tr>
<tr><td>Ours</td><td><strong>11.290</strong></td><td><strong>0.919</strong></td><td><strong>0.402</strong></td><td><strong>2.670</strong></td><td><strong>0.213</strong></td></tr>
</table>

We compare against He *et al.* [17] and StarGAN [21] by adapting their approaches to our setting of using face input images and allowing for continuous value gaze and head orientation redirection. He *et al.* [17] is the state-of-the-art in gaze redirection methods and is more accurate and photorealistic than previous approaches such as DeepWarp [15] and does not require synthetic training data as in Yu *et al.* [16]. StarGAN [21] is similar in approach to He *et al.* but is a more generic approach and thus important to compare to. We re-implement all baselines with a DenseNet-based architecture similar to ours to yield comparable results (implementation details are in the supplementary).

Our method outperforms the baseline methods, as measured by both the redirection and disentanglement metrics on four different datasets (see Tab. 2). We achieve precise and stricter control over gaze and head orientation thanks to the clear separation of task-relevant explicit factors and self-learned extraneous factors. The latter encode task-irrelevant changes which are disentangled from the explicit factors. As can be seen from Fig. 2, our method can generate photo-realistic results even for subjects with glasses and under large head orientation changes. He *et al.* [17] show periodic artifacts on faces and backgrounds. This may be due to the fact that only a perceptual loss is used as the reconstruction objective (instead of a pixel-wise L1/L2 loss), resulting in a constraint that is applied at lower image scales than the original resolution, which leads to such artifacts. StarGAN [21] maintains personal appearances well under small head orientation changes, but degenerates quickly with larger redirection angles. These comparisons show the importance of aligning extraneous factors between images during training which allows for stricter supervision on the generated images. Furthermore, our method can optionally match the lighting conditions of the target image by transforming the extraneous latent factors, as can be seen from columns (d) and (e). Importantly, our method more faithfully reproduces the target gaze direction and head orientation as shown in both qualitative and quantitative results. For more qualitative results, please refer to the supplementary materials.

## 4.5 Semi-supervised Cross-dataset Gaze Estimation

The value of learning to robustly isolate and control explicit factors from in-the-wild data lies in its potential to improve performance of downstream computer vision tasks, such as in training gaze or head orientation estimation models. Therefore, we perform experiments on semi-supervised

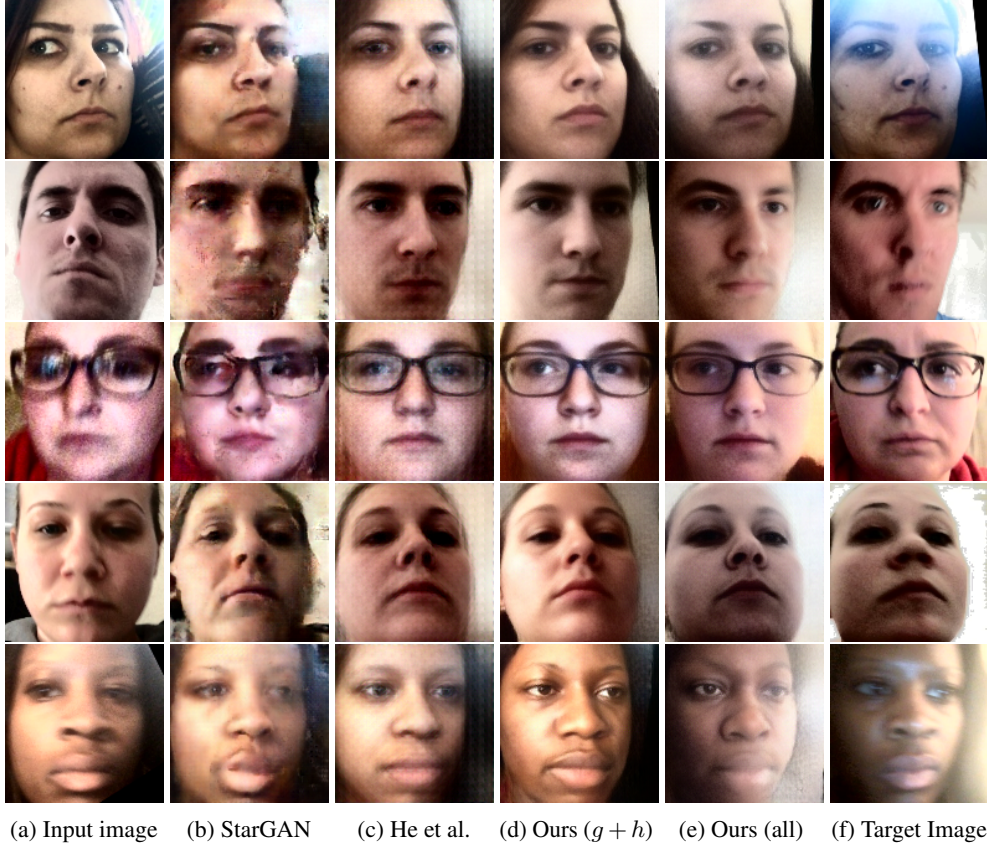

| (a) Input image | (b) StarGAN | (c) He et al. | (d) Ours $(g+h)$ | (e) Ours (all) | (f) Target Image |

Figure 2: **Qualitative Results.** Our method produces more detailed and photo-realistic images compared to the baseline methods of He *et al.* [17] and StarGAN [21]. By aligning to all predicted pseudo-labels of a target ground-truth image, our approach can also better approximate the target.

person-independent cross-dataset estimation on four popular gaze datasets. We show that even with small amounts of training data, our gaze redirector can extract and understand the variation of the dataset's factors sufficiently to augment it with new samples without introducing errors.

We randomly select $x \in \{2.5k, 5k, 10k, 20k, 50k\}$ labeled samples from the GazeCapture [18] training split and use the rest *without* labels to train ST-ED. While training, we employ pseudo label estimation loss $\mathcal{L}_{\mathrm{PL}}$ only for labeled images, and the $F_d$ used for functional loss $\mathcal{L}_{\mathrm{F}}$ is also pretrained on only the labeled subset. In forming our batches, we select only labeled samples for $X_i$ while $X_t$ can be either a labeled or unlabeled sample from the same person as $X_i$.

Once ST-ED is trained, we estimate the joint probability distribution function of the gaze and head orientation values of the labeled subset and sample random target conditions from it. We redirect the images in the labeled subset to these target conditions using ST-ED, and add them to the labeled set to create an augmented dataset (that is 2x the size of the labeled set). Lastly, we train a new gaze and head orientation estimation network (with the same architecture as $F_d$), but with this augmented set and compare its performance to the "baseline" version trained only with the smaller labeled dataset.

Fig. 3 shows that the gaze and head orientation estimation networks trained with both labeled data and augmented data (via redirection with a semi-supervised ST-ED) yields consistently improved performance on all four evaluation datasets. This is particularly true for cases with very few labeled samples (2,500), where the largest gains in performance are found. Approximately speaking, our method requires half the amount of labeled data to achieve similar performance as the baseline model. Importantly in our setting, our redirection network is not trained on any additional labeled data. This is in contrast to prior works which use additional synthetic [16] or real-world data [17] in the training of the redirector. Hence, this setting is very challenging, and to the best of our knowledge, we are the first to tackle this problem, demonstrating consistent improvements in downstream performance on multiple real-world datasets.

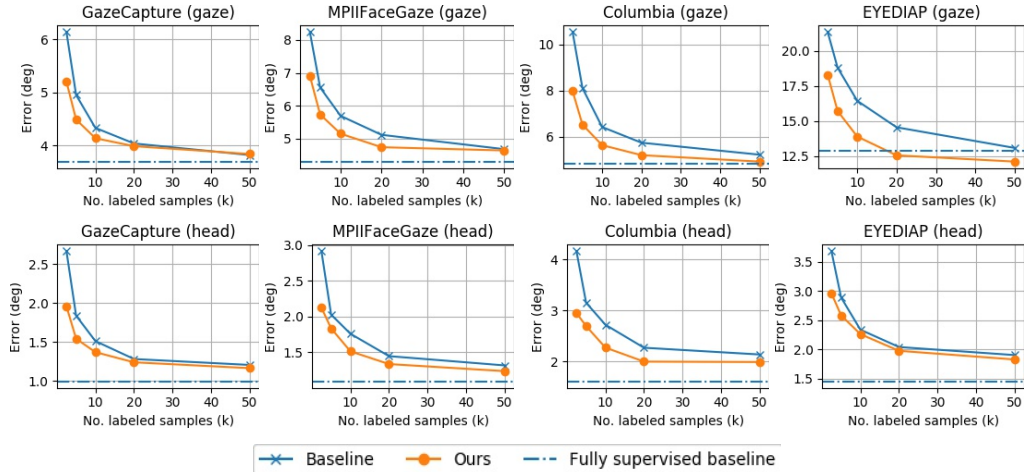

Figure 3: **Augmenting labeled training data via semi-supervised ST-ED.** We show that a semi-supervised learned ST-ED can augment an initial labeled dataset via redirection, to yield consistent improvements in cross-dataset gaze estimation benchmarks. Our method is consistently better than the baseline estimator trained on the initial labeled data only. This holds true for both gaze direction and head orientation estimation.

## 5 Discussion

We have discussed our method in the context of gaze and head-orientation estimation. However, we note that the architecture is motivated in a general sense and thus we are optimistic in its potential application to other conditional image generation tasks for which a small subset of labels is available and many more factors of variation need to be identified and separated without explicit supervision. We can discover and match the mis-alignment of extraneous variations with our self-learned transformations, and thus improve the learning of task-relevant factors. Furthermore, our proposed functional loss punishes perceptual differences between images with an emphasis on task-relevant features, which can be useful for various problems with an image reconstruction objective, for e.g., auto-encoding, neural rendering, etc. We leave further exploration of different application domains for future work.

## Broader Impact

Our work can perform accurate and photo-realistic gaze and head orientation redirection which makes augmenting existing datasets possible. It can also be used for film post-editing, group photo editing and video conferencing to correct the gaze directions and head orientations. We believe the method may have applications in other problem settings and thus it should be possible to leverage it to generate training data for estimators beyond gaze. Given that the method can generate realistic looking images under fine-grained control of selected parameters, it could also be leveraged for malicious manipulation of imagery in the context of "deep-fakes". Due to the limitations of currently available gaze datasets, our method does not yet handle extreme gaze directions that are beyond the distribution of the training dataset, nor fully faithfully preserve person-specific details in its redirection output. However, future developments should keep the ethical and privacy concerns in mind when refining such technologies.

## Acknowledgments and Disclosure of Funding

This project has received funding from the European Research Council (ERC) under the European Union's Horizon 2020 research and innovation programme grant agreement No. StG-2016-717054.

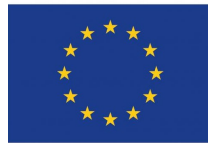 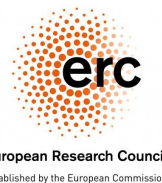

## Footnotes

[1]This "rotational equivariance" is enforced for a factor by assuming that the apparent difference in rotations in two images as defined by their ground-truth labels must apply equivalently at the latent-code level. That is, if the ground-truth between samples $X_i$ and $X_t$ suggest that there is a gaze direction difference of $30°$, this difference can be directly applied to the relevant latent code (via rotation matrices).

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
