[Supplementary Material]

# Self-Learning Transformations for Improving Gaze and Head Redirection

## (Supplementary Materials)

**Yufeng Zheng**[1], **Seonwook Park**[1], **Xucong Zhang**[1], **Shalini De Mello**[2], **Otmar Hilliges**[1]

[1]Department of Computer Science, ETH Zurich        [2]NVIDIA

{firstname.lastname}@inf.ethz.ch  shalinig@nvidia.com

## 1 Overview

In this supplementary document, we first show additional qualitative examples and experimental results (Sec. 2). We then provide details of the transformation function $T(\cdot)$ (Sec. 3) and implementation of network architectures (Sec. 4). We highly recommend that our readers view the supplementary video which provides further results produced by the proposed method.

## 2 Further Results

### 2.1 Video Samples

Please check the accompanying video for samples which further demonstrate the quality and consistency of our approach. Note that all samples are produced using a test subset of the GazeCapture dataset [1] and as such no over-fitted results are shown.

Our ST-ED approach is able to reconstruct smoother and more plausible changes in gaze direction and head orientation, and generates images with photo-realism despite being trained on a highly noisy dataset of in-the-wild images. Furthermore, the gaze direction and head orientation apparent in the output video sequences more faithfully reflect the given inputs, with promising results at extreme angles which go beyond the range of the training dataset (as such, the faithfulness of those generated samples cannot be quantitatively measured yet).

### 2.2 Additional Image Samples

In Fig. 2 (see end of document), we show additional qualitative results from the GazeCapture test set, comparing our method against state-of-the-art baselines.

### 2.3 Failure Cases

In Fig. 1 we show that our method sometimes exhibits difficulties in handling eyeglasses and expressions (Fig. 1a), preserving person-specific appearance characteristics such as face shape (Fig. 1b), or retaining finer details of the face such as moles and freckles (Fig. 1c).

### 2.4 Experiments on loss weight combinations

In Table 1, we show that our method is robust to different wights for the various loss terms. The fluctuation in redirection score is around 0.1 degrees when decreasing reconstruction, functional or pseudo label loss weights by half (compared to the final weights that we selected and show in Sec. 3.3 of the main paper). We also experiment with varying the weights of our embedding loss terms, which

Generated Target | Generated Target | Generated Target

(a) Eyeglasses and Expressions | (b) Person-specific face shape | (c) Finer details (freckles)

Figure 1: **Failure cases and limitations.** Most of our failure cases are related to appearance preservation. Difficult scenarios includes eyeglasses, unusual facial features or expressions and finer details such as freckles.

Table 1: **Sensitivity to loss term weights.** Our method is robust to different loss term weight combinations. The metrics remain reasonably consistent when the weight of a loss term is decreased by half or set to zero (for embedding losses).

| Approach | Gaze Direction | | | Head Orientation | | | LPIPS | |
|---|---|---|---|---|---|---|---|---|
| | Re-dir. | $u \to g$ | $h \to g$ | Re-dir. | $u \to h$ | $g \to h$ | $g + h$ | all |
| 0.5x weight for $\mathcal{L}_{\mathrm{R}}$ | 2.231 | 0.517 | **2.003** | **0.814** | 0.209 | 0.390 | 0.247 | 0.204 |
| 0.5x weight for $\mathcal{L}_{\mathrm{F}}$ | 2.321 | 0.558 | 2.225 | 0.831 | 0.227 | 0.397 | 0.253 | 0.206 |
| 0.5x weight for $\mathcal{L}_{\mathrm{PL}}$ | 2.249 | 0.550 | 2.070 | 0.819 | 0.229 | 0.411 | 0.248 | 0.204 |
| No embedding losses | 2.359 | **0.470** | 2.013 | 0.779 | **0.198** | 0.404 | **0.243** | **0.200** |
| Ours | **2.195** | 0.507 | 2.072 | 0.816 | 0.211 | 0.388 | 0.248 | 0.204 |

includes the embedding consistency loss $\mathcal{L}_{\mathrm{EC}}$ and the embedding part of our disentanglement loss $\mathcal{L}_{\mathrm{D}}$ (the last term in Eq. 9 of the main paper). Removing the embedding-related loss terms leads to slightly better LPIPS metric but worsened gaze redirection metric. Overall, we conclude that our loss combination is stable and robust to different weight values.

## 2.5 Comparison with He *et al.* [2] on the Downstream Task

We compare our method with baseline methods on the tasks of semi-supervised cross-dataset gaze and head pose estimation in Sec. 4.5 of the main paper. In this section, we further compare it with the previous state-of-the-art method from He *et al.* [2]. We first train the gaze redirector of the method from He *et al.* [2] with the *entire* GazeCapture training set, and then generate eye images from $2.5k$ real samples. Finally we train the estimator $F_d$ with both the real and generated data. Please note that our method as well as the baseline follow the same procedure where only the $2.5k$ real samples are used during the entire training procedure.

As shown in Table 2, augmenting the real data samples using the method from He *et al.* [2] generally results in performance degradation compared to the baseline, despite having used more labeled data for training the redirection network. This difference in implementation was necessary, as with very

Table 2: Downstream estimation error with $2.5k$ real training samples. We show the results from He *et al.* [2], a supervised baseline method and our method for both gaze and head pose estimation tasks. The redirector is trained on the whole GazeCapture training set for He *et al.* [2], and only $2.5k$ real samples for our method (in a semi-supervised fashion with unlabeled samples from the rest of the dataset).

(a) Gaze Direction

| Method | GazeCapture | MPIIGaze | ColumbiaGaze | EYEDIAP |
|---|---|---|---|---|
| He *et al.* [2] | 9.882 | 11.985 | 9.274 | 25.43 |
| Baseline | 6.138 | 8.243 | 10.536 | 21.35 |
| Ours | **5.203** | **6.903** | **7.974** | **18.31** |

(b) Head Orientation

| Method | GazeCapture | MPIIGaze | ColumbiaGaze | EYEDIAP |
|---|---|---|---|---|
| He *et al.* [2] | 13.457 | 7.964 | 3.787 | 13.04 |
| Baseline | 2.668 | 2.913 | 4.158 | 3.681 |
| Ours | **1.961** | **2.122** | **2.950** | **2.960** |

few samples the redirection network of He *et al.* could not be trained successfully. Furthermore, the approach of He *et al.* [2] cannot be trained in a semi-supervised manner.

This result highlights that smaller differences in redirector performance can cause large differences in downstream regression tasks. We believe that the degradation in performance is also due to the lower quality of the generated images from He *et al.* [2], which contain many artifacts as shown in Fig. 2. This causes a domain shift problem between the training images (half of which are generated images) and the testing images (real), which harms performance.

In contrast, our semi-supervised re-director is trained without any additional labeled data. Nonetheless, it can generate accurate and photo-realistic samples, which consistently improves performance over the baseline estimator.

## 3 Definition of Transformations

In a typical transforming encoder-decoder architecture, the encoder predicts an embedding and this embedding is transformed via pre-defined transformation routines such as translations as shown in the initial transforming autoencoder architecture by Hinton *et al.* [3]. We follow the approach of Worrall *et al.* [4] and Park *et al.* [1] and define our transformations as rotations, which are easily invertible and are linear orthogonal transformations. This makes such transformations easy to control and to some extent, interpret.

For a given factor of variation $f_i^j$, we assume as described in Sec. 3.2 of the main paper that this factor is described by an embedding $z_i^j$ and a pseudo condition $\tilde{c}_i^j$. More specifically, assuming that a rotation matrix $R_i^j$ is associated with this factor and its variation, we define that the embedding predicted by the encoder $G_{enc}$ is written as,

$$z_i^j = R_i^j z_{i_{canon}}^j, \tag{1}$$

where $z_{i_{canon}}^j$ is the canonical representation associated with input $X_i$. Based on this assumption, we have:

$$\tilde{z}_t^j = T\left(z_i^j,\, c_i^j,\, c_t^j\right) \tag{2}$$

$$= R_t^j \left(R_i^j\right)^{-1} z_i^j, \tag{3}$$

where the rotation matrices $R_t^j$ and $R_i^j$ are computed from $\tilde{c}_t^j$ and $\tilde{c}_i^j$, respectively. By the definition of $SO(\cdot)$ rotation matrices, the inverse of a given matrix is simply its transpose.

More specifically, at the stage of configuring the ST-ED architecture (introduced in Sec. 3.1 of the main paper), an arbitrary number of factors can be defined with $f = \left\{f^1,\, f^2,\, \ldots,\, f^N\right\}$ where each factor $f^j$ can be controlled with degrees of freedom $\in \{0,\, 1,\, 2\}$.

**0-DoF Factors.** For the case of zero degrees of freedom, we define $z_i^0$, which does not vary but is simply passed to the decoder assuming and enforcing that $z_i^0 \simeq z_t^0$ via the reconstruction objective (Eq. 3 of main paper).

**1-DoF Factors.** For the case of 1 degree of freedom, we define the rotation matrix of factor $f_i^j$ as:

$$R_i^j = \begin{pmatrix} \cos c_i^j & -\sin c_i^j \\ \sin c_i^j & \cos c_i^j \end{pmatrix} \tag{4}$$

and the dimensionality of the associated embedding $z_i^j$ becomes $N_f^j \times 2$, where $N_f^j$ is the hyperparameter for defining the latent embedding size for this 1-dimensional factor.

**2-DoF Factors.** For the case of 2 degrees of freedom, we define the rotation matrix of factor $f_i^j$ as:

$$R_i^j = \begin{pmatrix} \cos \phi_i^j & 0 & \sin \phi_i^j \\ 0 & 1 & 0 \\ -\sin \phi_i^j & 0 & \cos \phi_i^j \end{pmatrix} \begin{pmatrix} 1 & 0 & 0 \\ 0 & \cos \theta_i^j & -\sin \theta_i^j \\ 0 & \sin \theta_i^j & \cos \theta_i^j, \end{pmatrix} \tag{5}$$

where we define the components of the 2-dimensional condition $c_i^j = \left(\theta_i^j,\, \phi_i^j\right)$ and the dimensionality of the associated embedding $z_i^j$ becomes $N_f^j \times 3$. This is in line with the definition of

Table 3: Architecture of the PatchGAN discriminator used to train ST-ED

| Nr. | layers / blocks |
|---|---|
| 0 | Conv2d(3, 64, kw=4, stride=2, pad=1, bias=True), LeakyReLU() |
| 1 | Conv2d(64, 128, kw=4, stride=2, pad=1, bias=False), BatchNorm2d(128), LeakyReLU() |
| 2 | Conv2d(128, 256, kw=4, stride=2, pad=1, bias=False), BatchNorm2d(256), LeakyReLU() |
| 3 | Conv2d(256, 512, kw=4, stride=1, pad=1, bias=False), BatchNorm2d(512), LeakyReLU() |
| 4 | Conv2d(512, 1, kw=4, stride=1, pad=1, bias=True) |

spherical coordinate systems for head orientation and gaze direction estimation [5], where the zero-representation should correspond with a "frontal" direction, such as the face being oriented to be directly facing the camera.

# 4 Further Implementation Details

In this section, we provide further details of the configuration and implementation of our ST-ED, and the external regression networks $F_d$ and $F_d'$ for gaze estimation and head orientation. The codebase for this project can be found at https://github.com/zhengyuf/ST-ED

## 4.1 Self-Transforming Encoder-Decoder (ST-ED)

The ST-ED architecture can be flexibly configured. Here, we provide details of the backbone architecture used and the specific explicit and extraneous factors that were configured, along with their latent embedding dimensions. Lastly, we provide the used hyperparameters during training for better reproducibility.

### 4.1.1 Network Architecture

**Generator** We use the DenseNet architecture to parameterize our encoder and decoder [6]. For our decoder, we replace the convolutional layers with de-convolutions and the average-pooling layers with strided $3 \times 3$ de-convolutions. We configure the DenseNet with a growth rate of 32. Our input image size is $128 \times 128$, and we use 5 DenseNet blocks and a compression factor of 1.0. We don't use dropout or $1 \times 1$ convolutional layers, and use instance normalization and leaky ReLU. The feature map size at the bottleneck is $2 \times 2$, and we flatten the features and pass them through fully-connected layers to calculate the embeddings and pseudo-labels. Before decoding, we reshape the embeddings to have a spatial resolution of $2 \times 2$ to match the bottleneck's shape.

In all our experiments, we use a 0-DoF factor of size 1024, $4\times$ 1-DoF factors of size $16 \times 2$ and $4\times$ 2-DoF factors of size $16 \times 3$ for our generator. Two of the 2-DoF factors are chosen to represent the explicit factors, i.e., gaze direction and head orientation.

**Discriminator** We use a PatchGAN discriminator as in [7]. The receptive field at the output layer is $70 \times 70$, and the output size is $14 \times 14$. The architecture of the discriminator is listed in Tab. 3.

### 4.1.2 Training Hyperparameters

We use a batch size of 20, and train the network for 3 epochs (about 210k iterations). The initial learning rate is $10^{-3}$ and is decayed by $0.8$ every $0.5$ epoch. We use the Adam optimizer [8] with a weight decay coefficient of $10^{-4}$. We use the default momentum value of $\beta_1 = 0.9$, $\beta_2 = 0.999$

## 4.2 Gaze Estimation and Head Orientation Network

We use a VGG-16 [9] network to implement $F_d$. We select an ImageNet [10] pre-trained model and fine-tune it on the gaze and head orientation estimation tasks. The input to this network is a full-face image of size $128 \times 128$ pixels, and the output is a 4-dimensional vector representing pitch and yaw values for gaze direction and head orientation. The architecture of the network is shown in Tab. 5.

Table 4: Architecture of the external gaze direction and head orientation estimation network, $F_d$.

| Nr. | layers / blocks |
|---|---|
| 0 | VGG-16 convolutional layers |
| 1 | FC(512, 64, w/bias), LeakyReLU() |
| 2 | FC(64, 64, w/bias), LeakyReLU() |
| 3 | FC(64,4, w/bias), $0.5\pi\cdot$Tanh() |

Table 5: Architecture of the external gaze direction and head orientation estimation network, $F_d'$.

| Nr. | layers / blocks |
|---|---|
| 0 | ResNet convolutional layers, stride of MaxPool2d = 1 |
| 1 | FC(2048, 4, w/bias) |

We fine-tune the network for $100k$ iterations with a batch size of 64, using the Adam optimizer[8] with momentum values $\beta_1 = 0.9$, $\beta_2 = 0.95$. The initial learning rate is $10^{-4}$ and is decayed by a factor of 0.5 after $50k$ iterations.

The external estimator for evaluation $F_d'$ is trained in a similar way as $F_d$, but with a ResNet-50 [11] backbone which is also pre-trained on ImageNet.

### 4.3 State-of-the-Art Baselines

There exists no prior art in simultaneous head and gaze redirection and as such we extend and re-implement the state-of-the-art approach for gaze redirection, He *et al.* [2] and its close-cousin, StarGAN [12]. We perform this as best as possible by being faithful to the original objective formulations, but implement a backbone similar to our own ST-ED for fairness. This sub-section provides further details of our implementation.

#### 4.3.1 He *et al.*

This work originally uses eye images ($64 \times 64$) from the Columbia Gaze dataset [13], and performs only gaze redirection. No head orientation manipulation was shown. To apply the method to our dataset and to ensure a fair comparison, we parametrize the generator with a DenseNet-based encoder-decoder architecture, which is conceptually similar to the original down- and up-sampling generator from He *et al.* [2]. We use one fewer DenseNet block compared to our ST-ED approach in both the down- and up-sampling stages of the generator, in order to match the original implementation of He *et al.* [2] which has a spatially wider bottleneck than our implementation of ST-ED. To extend the work of He *et al.* [2] to perform both gaze and head redirection, we simply estimate both values with the estimation branch of the discriminator, and optimize for both.

We use a global discriminator, as in the original implementation from He *et al.* [2]. The discriminator predicts a 5-dimensional vector representing the discriminator value, gaze direction and head orientation. We use a Tanh() function on the gaze and head direction values and then multiply by $0.5\pi$ to match the ranges of the pitch and yaw values. We follow the original implementation from He *et al.* [2] and use the WGAN-GP objective [14]. The discriminator architecture is given in Tab. 6.

We increase the weight for gaze (and head orientation) estimation loss of He *et al.* [2] from 5 to 1000, because gaze is harder to estimate when using full face images and the training estimation loss fails to converge when using the original weight of 5. The weights for the other loss terms are the same as in the original implementation of He *et al.* [2]. We choose the same training hyper-parameters as in our method, except that we train the network for 6 instead of 3 epochs since the WGAN-GP objective updates the generator less frequently.

#### 4.3.2 StarGAN

Our implementation of StarGAN [12] uses the same generator and discriminator architecture as our re-implementation of He *et al.* [2]. Since StarGAN does not need paired training images, we train it by redirecting the input images to random gaze and head directions, which are sampled from a 4D

Table 6: Global discriminator network with a regression branch for gaze direction and head orientation, as used in the re-implementation of the He *et al.* [2] and StarGAN [12] approaches.

| Nr. | layers / blocks |
|-----|-----------------|
| 0 | Conv2d(3, 64, kw=4, stride=2, pad=1, bias=True), LeakyReLU() |
| 1 | Conv2d(64, 128, kw=4, stride=2, pad=1, bias=False), LeakyReLU() |
| 2 | Conv2d(128, 256, kw=4, stride=2, pad=1, bias=False), LeakyReLU() |
| 3 | Conv2d(256, 512, kw=4, stride=2, pad=1, bias=False), LeakyReLU() |
| 3 | Conv2d(512, 1024, kw=4, stride=2, pad=1, bias=False), LeakyReLU() |
| 3 | Conv2d(1024, 2048, kw=4, stride=2, pad=1, bias=False), LeakyReLU() |
| 3 | Conv2d(2048, 5, kw=2, stride=1, pad=0, bias=False) |

joint distribution of gaze and head directions computed by fitting a Gaussian kernel density to the ground truth labels from the training dataset. The weights for the cycle consistency, GAN, and gaze and head estimation losses are 400, 1 and 1000, respectively.

## 4.4 Data Pre-processing

We preprocess our image data in the same manner as done in Park *et al.* [1], but with changes to yield face images. That is, we follow the pipeline defined by Zhang *et al.* [5] and originally proposed by Sugano *et al.* [15] but select [16] and [17] respectively for face detection and facial landmarks localization. We use the Surrey Face Model [18] and the same reference 3D landmarks as in [1] to perform the data normalization procedure. To yield $128 \times 128$ images, we configure our virtual camera to have a focal length of $500mm$ and distance-to-subject of $600mm$.

In more simple terms, we use the code[1] provided by Park *et al.* [1] and tweak the parameters in the "normalized_camera" variable.

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

Figure 2: continued on the next page...

(a) Input image  (b)  StarGAN [12]  (c) He et al. [2]  (d) Ours $(g+h)$  (e) Ours (all)  (f) Target Image

Figure 2: **Qualitative Results.** Example redirection results on the test subset of GazeCapture [19]. Our method produces more detailed and photo-realistic images compared to the baseline methods of He *et al.* [2] and StarGAN [12]. Note that our method can generate photo-realistic images even in cases of large head pose changes, eyeglasses, and blurry inputs. By aligning to all predicted pseudo-labels of a target ground-truth image, our approach can also better approximate the target.