[Reviews · NeurIPS 2020]

Review 1

Summary and Contributions: The paper introduces a novel method for gaze redirection. Furthermore, authors use the proposed method to train gaze prediction models with fewer labels, showing the ability of their model to augment the data. Authors evaluate their method on various standard benchmarks and show how their model improves existing models in the redirection tasks.

Strengths: - The paper describes the model and the task very well, and justify the existence of the different part of the model. - Authors show how the model improves existing models on gaze and head redirection tasks in multiple dataset. - I believe that tackling explicitly the head/gaze difference is an interesting technical addition to the model. - The augmentation experiments show a very interesting use of the model for data augmentation in a low data regime. I think the fact the authors are able to improve baseline performance in a low data regime is interesting and useful for future research. Furthermore, the fact that they can control gaze direction enables new training schemes which weren't possible with static datasets.

Weaknesses: - Authors address the possibility of using this work to build deep fakers in the broader impact statement. Although their point is valid, the model for now only works on particular poses, I still think this line of research help the overall development of fake videos. I think this should be taken more into account in the discussion in the paper. - Authors claim in the conclusion that the model would apply to other factors different from head pose and gaze in generation. Although the model is general, I think every problem is very particular and the claim might be too strong. - It would have been interesting to train the model with other datasets to see how the performance change with training data. Authors do claim that their model transfers very well across datasets (which is true), but it would be interesting to see if it still does it with other training data. - How did the authors validate how well F_d works and how sensible is to small changes? It is used to evaluate the effect of disturbance and I believe it is importance to also evaluate the evaluator itself. Otherwise, the metrics involving F_d are hard to interpret for the reader. After rebuttal: After reading the rebuttal, the authors have addressed most of my concerns and I update my score to 7-Accept.

Correctness: Yes. The claims on the paper are well formulated. The method seems correct and well supported by the empirical results, both qualitative and quantitative.

Clarity: Yes, the paper is well organised and well written.

Relation to Prior Work: Yes, the authors do describe in detail the other works attempting to do gaze redirection, as well as describe related work on generation and gaze estimation.

Reproducibility: Yes

Additional Feedback:


Review 2

Summary and Contributions: In this paper, a new generative model for producing high-quality face images of eye gaze and head pose redirected. For this a new encoder-decoder architecture that disentangles various independent factors of faces. Several constraints were designed for the disentanglement and redirection. The newly proposed method was validated by three error measures and further shown its improvements in semi-supervised cross-dataset gaze estimation.

Strengths: This paper proposes a new structure that is simple and works well. The proposed method is clearly described and explained, and it shows the superiority of its performance through various experiments. The performances (especially qualitative results) are clearly better in many ways compared to the existing comparative methods.

Weaknesses: It is unlikely that it will be easy to balance the six losses in use in the proposed method (i.e. parameter setting). More specific discussions and experiments related to this parameter setting are needed. It would be nice to have a frank discussion of the proposed method's limitations by showing the failure cases.

Correctness: Yes, they seem correct.

Clarity: The paper was well written and easy to follow. The overall structure is also logical, and the motivations of the research and the issues being addressed are clearly described.

Relation to Prior Work: Yes, analysis and comparison of other papers have been done sufficiently.

Reproducibility: Yes

Additional Feedback: It would be better if there was a comparison with FAZE [20] in the experiment.


Review 3

Summary and Contributions: The paper proposes a method for re-targeting/redirecting eye gaze and head pose. It specifically targets lower quality images. Further, it demonstrates the usefulness of such a method outside directly using it for redirection, but also as a way to augment training data for eye gaze estimation tasks.

Strengths: The paper shows impressive qualitative results on the task of gaze and head redirection. Especially impressive is the temporal stability of redirection of both eye gaze and head pose. The authors present an interesting way to evaluate their work, by augmenting the training data with their redirection network and training a downstream task. This is an interesting and good way to evaluate various redirection/controllable image generation work.

Weaknesses: - The paper method section is a bit difficult to follow (see below) - There is a potential issue in evaluation (see below) - It would be great to include a limitation section discussing where the approach still struggles

Correctness: There is one potential issue with evaluation in terms of redirection error (F_d). Is this the same F_d as is used in the loss function? If so, it is unsurprising it goes down as the optimized is told to reduce it. The model may be learning to exploit the biases of F_d rather than learning to redirect gaze. Ideally, you would want to use a different method to estimate gaze that the one present in the loss function.

Clarity: The introduction/background and evaluation sections are clearly written. However, the method section is difficult to follow at times, and required several re-reads to understand, some specifically unclear parts: 1. It is not entirely clear what part the Figure 1 notation with curved arrow in a circle alludes to. The method description does not have \delta c in it, is it Equation 2? 2. Relationship between factors and conditions is not immediately clear. Why do we need such separation. Is there a deterministic mapping between each embedding and a condition? 3. Is condition a scalar? If so, how can gaze (2-dimensional) and head pose (3-dimensional) be encoded in a condition? 4. It is never explained what is meant by rotationally equivariant mappings 5. [minor] Figure 1 could do with subfigures (rather than stating top left etc.)

Relation to Prior Work: Prior work and differences to it are clearly explained.

Reproducibility: Yes

Additional Feedback: Some relevant work that authors might be interested in "CONFIG: Controllable Neural Face Image Generation". It is completely understandable that the work was not cited as it come out very recently, but authors might find some similarities to their work. Typos: highly-quality -> high-quality

[Author Response · NeurIPS 2020]



We thank the reviewers for noting that our approach is novel and justified (R1), simple and clearly described (R2), interesting (R1, R3), and that our qualitative results are impressive (R2, R3). We will clarify any open points below.

**1) Use of $F_d$ for evaluation (R3)**. To analyze the effect of using identical $F_d$ models in training and in evaluation, we train a separate ResNet-50 on GazeCapture (training set) to use only for evaluations (results in the tables below). Even with the separate $F_d$ our approach outperforms state-of-the-art methods. This is in line with the trend shown in Tab. 2 of the main paper. We will update the table accordingly in the camera-ready.

| | GazeCapture | | | | MPIIFaceGaze | | | | Columbia | | | | EYEDIAP | | | |
|---|---|---|---|---|---|---|---|---|---|---|---|---|---|---|---|---|
| | Gaze Redir. | Head Redir. | $g \to h$ | $h \to g$ | Gaze Redir. | Head Redir. | $g \to h$ | $h \to g$ | Gaze Redir. | Head Redir. | $g \to h$ | $h \to g$ | Gaze Redir. | Head Redir. | $g \to h$ | $h \to g$ |
| StarGAN | 4.602 | 3.989 | 0.755 | 3.067 | 4.488 | 3.031 | 0.786 | 2.783 | 6.522 | 3.444 | 1.029 | 3.359 | 14.906 | 3.929 | 0.915 | 4.025 |
| He *et al.* | 4.617 | 1.392 | 0.560 | 3.925 | 5.092 | 1.372 | 0.684 | 3.411 | 7.345 | 1.677 | 0.692 | 3.831 | 13.548 | 1.581 | 0.663 | 4.367 |
| Ours | **2.195** | **0.816** | **0.388** | **2.072** | **2.233** | **0.884** | **0.365** | **1.849** | **3.333** | **1.095** | **0.452** | **2.136** | **11.290** | **0.919** | **0.402** | **2.670** |

**2) How well $F_d$ works (R1)**. The histogram to the right plots the differences between $F_d$ gaze angle predictions, for randomly sampled image pairs from the GazeCapture test set, against the corresponding ground-truth deltas. The plot indicates a strong correlation between the output from $F_d$ and the ground-truth differences (Pearson corr. coeff. of 0.92). This provides evidence that $F_d$ is a useful choice as an evaluation network and that it can reliably assess changes in gaze direction. We will add this analysis to the camera-ready.

**3) Training on other datasets (R1)**. We train our approach on MPIIFaceGaze and tabulate the cross-dataset results below (similar to Tab. 2 of the main paper). As expected, the performance decreases with MPIIGaze which has much fewer (15) subjects v.s. GazeCapture (993 subjects after pre-processing). However, our performance remains consistently better than state-of-the-art methods across all test datasets.

| | GazeCapture | | | | | Columbia | | | | | EYEDIAP | | | | |
|---|---|---|---|---|---|---|---|---|---|---|---|---|---|---|---|
| | Gaze Redir. | Head Redir. | $g \to h$ | $h \to g$ | LPIPS | Gaze Redir. | Head Redir. | $g \to h$ | $h \to g$ | LPIPS | Gaze Redir. | Head Redir. | $g \to h$ | $h \to g$ | LPIPS |
| StarGAN | 5.684 | 7.093 | 0.778 | 2.906 | 0.324 | 7.503 | 8.031 | 0.936 | 2.596 | 0.469 | 15.194 | 7.591 | 0.772 | 2.741 | 0.468 |
| He *et al.* | 5.788 | 2.874 | 0.755 | 5.064 | 0.299 | 8.156 | 4.031 | 0.946 | 5.197 | 0.469 | 16.904 | **3.283** | 0.696 | 6.005 | 0.407 |
| Ours | **3.064** | **2.764** | **0.391** | **1.821** | **0.261** | **3.955** | **3.833** | **0.405** | **1.625** | **0.424** | **14.624** | 3.423 | **0.308** | **1.648** | **0.362** |

**4) Balancing of loss terms (R2)**. While we did not explore all potential combinations of loss term weights, we found that our method is generally robust to how the weights are specifically set. This is evident from the coarse values chosen (and written in line 192 of our submission). In general, balancing the weighted contributions of the sub-objectives given the magnitude of their raw values is a good guideline. Additionally, an emphasis on the reconstruction loss term improves training stability, and a higher functional loss term weight leads to better redirection accuracy. We will provide some experimental results regarding these observations in our final version.

**5) Limitations & failure cases (R2, R3)**. We notice at times that there is difficulty in handling eyeglasses, and that person-specific appearance characteristics (e.g. face shape) are not always preserved fully. In addition, finer details of the face such as moles and freckles are not retained in the output. See 02:45 of our supplementary video submission for an example with eyeglasses. We will expand our discussion of limitations in the final version.

**6) Comparison with FAZE (R2)**. Tab. 1 of our supplementary text shows results of the base model without pseudo-label prediction at the output of the encoder, i.e., the ground-truth is used to rotate the predicted embeddings (as is done in transforming encoder-decoder architectures). This is equivalent to FAZE + a discriminator – we use the same backbone network as FAZE. The addition of the discriminator alone does not improve FAZE. Note that FAZE itself was not proposed as a gaze redirection method and performs poorly, producing images of very low quality.

**7) Explanation of factors and conditions in ST-ED (R3)**. Each factor is composed of an embedding ($z$) and condition ($c$) where the condition describes the amount of variation of the embedding (via rotations). This is defined by the transforming encoder-decoder framework where the encoder predicts a rotated embedding, and the decoder takes as input the same embedding, but with a different rotation. As further discussed in the supplementary text (Sec. 3), the conditions can be 1 or 2-dimensional. Head orientations are 2-dimensional in our setting (line 198 of the main paper).

**8) Miscellaneous (R1, R3)**. We will extend the discussion on the relation of our work to deep fakes, and tone down our claim regarding the generalization possibility of ST-ED to other CV tasks (R1). We will improve Fig. 1 as per suggestions, explain the concept of rotational equivariance more clearly, and cite the CONFIG paper (R3).

[Meta-Review · NeurIPS 2020]

All reviewers are impressed with both quantitative and qualitative results in the paper. They also appreciate the simplicity of the method and the clarity of the writing. This makes for solid grounds for acceptance -- I am however recommending a poster since this is a somewhat niche area.